# Lifted TASEP: long-time dynamics, generalizations, and continuum limit

**Fabian H.L. Essler**[1*]**, Jeanne Gipouloux**[2] **and Werner Krauth**[2,1,3†]

**1** Rudolf Peierls Centre for Theoretical Physics, Clarendon Laboratory, Oxford OX1 3PU, UK
**2** Laboratoire de Physique de l'Ecole normale supérieure, ENS, Université PSL, CNRS, Sorbonne Université, Université de Paris Cité, 24 rue Lhomond, 75005 Paris, France
**3** Simons Center for Computational Physical Chemistry, New York University, New York (NY), USA

⋆ fab@thphys.ox.ac.uk , † werner.krauth@ens.fr

## Abstract

We investigate the lifted TASEP and its generalization, the GL-TASEP. We analyze the spectral properties of the transition matrix of the lifted TASEP using its Bethe ansatz solution, and use them to determine the scaling of the relaxation time (the inverse spectral gap) with particle number. The observed scaling with particle number was previously found to disagree with Monte Carlo simulations of the equilibrium autocorrelation times of the structure factor and of other large-scale density correlators for a particular value of the pullback $\alpha_{\text{crit}}$. We explain this discrepancy. We then construct the continuum limit of the lifted TASEP, which remains integrable, and connect it to the event-chain Monte Carlo algorithm. The critical pullback $\alpha_{\text{crit}}$ then equals the system pressure. We generalize the lifted TASEP to a large class of nearest-neighbour interactions, which lead to stationary states characterized by non-trivial Boltzmann distributions. By tuning the pullback parameter in the GL-TASEP to a particular value we can again achieve a polynomial speedup in the time required to converge to the steady state. We comment on the possible integrability of the GL-TASEP.

## 1 Introduction

In recent years, Monte Carlo algorithms based on non-reversible Markov chains have received a growing amount of attention. In several important applications [1–3], they empirically outperform reversible Monte Carlo algorithms built on the detailed-balance condition. The message of these works is that breaking reversibility can improve on the slow diffusive exploration of high-dimensional sample spaces.

The theoretical analysis of lifted [4, 5] non-reversible Markov chains is involved because typically the spectrum of the transition matrix is complex valued and (left or right) eigenvectors do not form an orthonormal basis [6, 7]. In order to bridge the gap between exactly solved lifted single-particle models [4, 5] and real-life applications [8], the study of non-reversible lifted Markov chains for interacting many-particle systems in one spatial dimension was initiated in Refs [9–11]. In continuum systems, this leads to implementations of the event-chain Monte Carlo algorithm [12, 13], while for lattice systems this connects lifted Markov chains to the vast literature on exactly solvable (both reversible and irreversible) Markov-chain models such as the asymmetric simple exclusion process [14–29].

In a recent work [30], two of us proposed the lifted TASEP as a paradigm for non-reversible *lifted* Markov chains in one-dimensional particle systems. The model considers $N$ hard-sphere particles on an $L$-site lattice with periodic boundary conditions with only a single particle

being active. It carries a pointer which allows it to move in forward direction or to undergo a collision. In a second part of the move, the pointer itself moves to the nearest neighbour to the left, which becomes the new active particle:

$$
\underbrace{\boxed{\bullet\;\overset{\bullet\!\!\to}{}\;\;\;\bullet}}_{x_t} \to \boxed{\bullet\;\;\overset{\bullet\!\!\to}{}\;\bullet} \to \begin{cases} \underbrace{\boxed{\overset{\bullet\!\!\to}{}\;\;\bullet\;\bullet}}_{} & \alpha \\ \underbrace{\boxed{\bullet\;\;\overset{\bullet\!\!\to}{}\;\bullet}}_{x_{t+1}} & 1-\alpha \end{cases}
\tag{1}
$$

$$
\underbrace{\boxed{\bullet\;\overset{\bullet\!\!\to}{}\;\bullet\;}}_{x_t} \to \boxed{\bullet\;\bullet\;\overset{\bullet\!\!\to}{}\;} \to \begin{cases} \boxed{\bullet\;\overset{\bullet\!\!\to}{}\;\bullet\;} & \alpha \\ \underbrace{\boxed{\bullet\;\bullet\;\overset{\bullet\!\!\to}{}\;}}_{x_{t+1}} & 1-\alpha \end{cases} .
\tag{2}
$$

For all $0 < \alpha < 1$, the lifted TASEP is irreducible and aperiodic [6,30], and the $\alpha$-independent steady state is the equal-probability mixture of all possible lifted configurations (which are characterized by the positions of particles and the pointer). In Ref. [30] it was shown that the lifted TASEP is exactly solvable by means of Bethe-like ansatz, which was then used to establish bounds of the scaling of the spectral gap $\Delta$ with particle number. The results reported in Ref. [30] suggested that, for $N, L \to \infty$ with $N/L$ fixed, one has

$$
\Delta = \mathrm{Re}\ln[E^*] \le \begin{cases} \mathrm{const}\;N^{-5/2} & \text{if } \alpha \neq \alpha_{\mathrm{crit}}\,, \\ \mathrm{const}\;N^{-2} & \text{if } \alpha = \alpha_{\mathrm{crit}}\,, \end{cases}
\tag{3}
$$

Here $\alpha_{\mathrm{crit}} = N/L$ and $E^* \neq 1$ is the eigenvalue of the transition matrix with magnitude closest to one. These results were complemented by numerical simulations of the integrated autocorrelation time of the equilibrium structure factor $\tau_{\mathrm{IAC}}$, which were compatible with

$$
\tau_{\mathrm{IAC}} \sim \begin{cases} N^{5/2} & \text{if } \alpha \neq \alpha_{\mathrm{crit}}\,, \\ N^{3/2} & \text{if } \alpha = \alpha_{\mathrm{crit}}\,. \end{cases}
$$

Interestingly, at the special value $\alpha = \alpha_{\mathrm{crit}}$, the lifted TASEP displays a polynomial speedup in its approach to the steady state, on top of the $\sim N^{1/2}$ speedup that the generic-$\alpha$ case achieves compared to the SSEP (the symmetric simple exclusion process, in other words the Metropolis algorithm). However, as noted in Ref. [30], the scalings of $\tau_{\mathrm{IAC}}$ and $\Delta$ do not agree at $\alpha = \alpha_{\mathrm{crit}}$. One of the main aims of the present work is to resolve this discrepancy.

The outline of this work is as follows. In Section 2, we expand on the analyses of both $\Delta$ and $\tau_{\mathrm{IAC}}$ as functions of particle number $N$ and the pullback parameter $\alpha$. By numerically solving the Bethe equations for particular families of eigenstates of the transition matrix and considering particle numbers up to $N \sim 500$ (compared to $N \sim 250$ in Ref. [30]) we are able to clearly exhibit the crossover (as a function of $\alpha$) between the asymptotic $N^{-5/2}$ scaling of $\Delta$ at $\alpha \neq \alpha_{\mathrm{crit}}$ and the $N^{-2}$ scaling at $\alpha = \alpha_{\mathrm{crit}}$. We observe the same crossover behaviour in Monte Carlo simulations of the integrated autocorrelation time of the structure factor. In Section 3 we turn to the discrepancy (see above) between the scaling behaviours in $\tau_{\mathrm{IAC}}$ and $\Delta$ at $\alpha = \alpha_{\mathrm{crit}}$. By carefully keeping track of the translational invariance of the problem we identify which observables are a priori sensitive to the eigenvector of the transition matrix that gives rise to the scaling of $\Delta$. We then show that for small particle numbers the contribution of the eigenvector of interest to dynamical susceptibilities is too small to be observed in Monte Carlo simulations. We propose that this smallness of the relevant matrix elements, combined with a small number of eigenvectors whose eigenvalues scale as $N^{-2}$, makes it essentially impossible to detect the asymptotic relaxation time numerically. In Section 4, we construct a continuum limit (in both space and time) of the lifted TASEP that remains integrable and derive the Bethe ansatz equations that determine the eigenvalues of the transition matrix. We discuss the equivalence of this continuum process with the hard-sphere event-chain Monte Carlo

algorithm. In particular, the pullback $\alpha$ is related to the pressure, as discussed previously, and the critical pullback $\alpha_{\text{crit}}$ is seen to correspond to a vanishing pressure [11]. In Section 5, we generalize the lifted TASEP to a wide class of nearest-neighbour interactions. The GL-TASEP by construction provides a lifted Markov chain whose stationary state is the targeted Boltzmann distribution of interest. We present some preliminary results on the possible integrability of the GL-TASEP. Finally, Section 6 contains our conclusions.

## 2 Lifted TASEP: relaxation and autocorrelation times

### 2.1 Monte Carlo computations of autocorrelation times

In order to determine the scaling of the relaxation time with particle number we have carried out extensive Monte Carlo simulations of autocorrelation functions. These start from a random configuration $x_0$ and under time evolution give rise to a trajectory $\{x_0, x_1, x_2, \dots\}$. For individual trajectories $x_t = \{j_1(t), \dots, j_N(t); p(t)\}$ where $j_n(t)$ and $p(t)$ denote respectively the particle and pointer positions we determine time-dependent observables $f_t = f(x_t)$, and use these to determine autocorrelation functions

$$C(t) = \langle f_s f_{s+t} \rangle - \mu^2. \tag{4}$$

Here, $\langle \dots \rangle$ denotes a sample average and $\mu = \langle f \rangle$. In practice, it is convenient to focus on integrated autocorrelation times of normalized autocorrelation functions [31], which are defined as

$$\tau_f = \frac{1}{2} \sum_{t=-\infty}^{\infty} \frac{C(t)}{C(0)}. \tag{5}$$

In the following, we present results for the structure factor $f_t = |S(2\pi/L, t)|^2$, defined as

$$|S(q,t)|^2 = \frac{1}{N} \left| \sum_{r=1}^{L} e^{iqr} \rho(r,t) \right|^2, \; \rho(r,t) = \sum_{n=1}^{N} \delta_{r,j_n(t)} \,. \tag{6}$$

This observable is sensitive to long-range density fluctuations, which are expected to relax slowly in equilibrium. We have analyzed a number of other observables in order to verify that the relaxational behaviour seen for the structure factor is generic. Examples are

$$S(q,t) = \frac{1}{\sqrt{L}} \sum_{r=1}^{L} e^{iqr} \rho(r,t) \,, \qquad \mathcal{O}_1(q,d) = \sum_{r,r'=1}^{L} e^{iqr} \rho(r,t) \rho(r',t) \delta_{|r-r'|,d} \,. \tag{7}$$

Our equilibrium Markov-chain Monte Carlo simulations, presented in Ref. [30], show quite clearly that

$$\tau_{\text{IAC}} \equiv \tau_{|S(q)|^2} \sim \begin{cases} N^{5/2} & \text{for } \alpha \neq \alpha_{\text{crit}}, \\ N^{3/2} & \text{for } \alpha = \alpha_{\text{crit}}. \end{cases} \tag{8}$$

For $\alpha \lesssim \alpha_{\text{crit}}$, the asymptotic $N^{5/2}$ scaling is reached only for large system sizes (see Fig. 1).

### 2.2 Bethe ansatz solution of the lifted TASEP

Configurations in the lifted TASEP are labelled by the positions $j_1 < \cdots < j_N$ of the $N$ particles and an integer $1 \leq a \leq N$, which identifies the pointer among the particles. In this basis, the eigenvectors of the transition matrix have amplitudes

$$\psi_a(\boldsymbol{j}) = \sum_{Q \in S_N} A_a(Q) \prod_{j=1}^{N} (z_j)^{j_{Q_j}} \,, \tag{9}$$

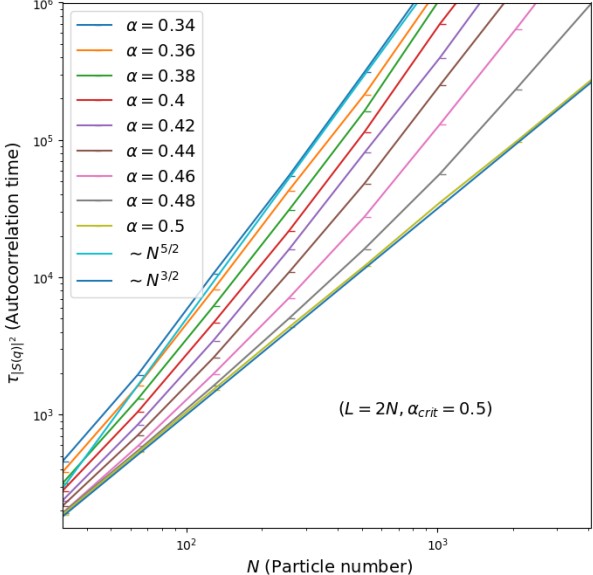

Figure 1: Autocorrelation functions of the structure factor at $L = 2N$ (with $\alpha_{\text{crit}} = 0.5$), for different values of the pullback $\alpha$. For $\alpha < \alpha_{\text{crit}}$, the asymptotic scaling is as $\sim N^{5/2}$, but for $\alpha \lesssim \alpha_{\text{crit}}$, this is reached only for very large $N$. The scalings $\sim N^{3/2}$ and $\sim N^{5/2}$ are indicated through straight lines.

where the rapidities $z_j$ and $E$ are solutions to the following set of coupled equations [30]

$$z_a^{L-1} = \frac{\frac{1-\alpha}{z_a} - E}{\alpha}(-1)^N \prod_{b=1}^N \frac{E - \alpha - \frac{1-\alpha}{z_b}}{E - \alpha - \frac{1-\alpha}{z_a}}, \quad a = 1, \dots, N,$$
$$\prod_{j=1}^N \left(E - \frac{1-\alpha}{z_j}\right) = \alpha^N \prod_{k=1}^N \frac{1}{z_k}. \tag{10}$$

Periodic boundary conditions imply that

$$\psi_N(j_1, \dots, j_{N-1}, L+1) = \psi_1(1, j_1, \dots, j_{N-1}),$$
$$\psi_1(0, j_2, \dots, j_N) = \psi_N(j_2, \dots, j_N, L). \tag{11}$$

The $L$-site lattice with periodic boundary conditions is translationally invariant and, therefore, the eigenvectors of the transition matrix all have definite momenta. The latter can be worked out by considering a translation by one site, and by observing that Eqs (9) and (11) imply that

$$\psi_a(j_1+1, \dots, j_N+1) = \left(\prod_{n=1}^N z_n\right) \psi_a(j_1, \dots, j_N). \tag{12}$$

We conclude that the momentum of a left eigenstate with rapidities $\{z_j\}$ is

$$P = i \sum_{n=1}^N \ln(z_n). \tag{13}$$

In the case $L = 2N$, we may reparametrize the Bethe ansatz equations using

$$\beta = \frac{\overline{\alpha}}{E - \alpha} \, , \quad u_a = \frac{2z_a}{\beta} - 1 \, , \quad \delta = \frac{\beta - \overline{\alpha}/\alpha}{\beta + \overline{\alpha}/\alpha} \, ,$$

$$\mu = \left( \frac{2}{\beta} \right)^L \frac{1 - \alpha(1 - \beta)}{2\alpha} \prod_{b=1}^{N} \frac{u_b - 1}{u_b + 1} \, . \tag{14}$$

This maps Eq. (10) onto

$$\left( 1 - u_a^2 \right)^{\frac{L}{2}} = -\mu(u_a + \delta) \, , \quad a = 1, \ldots, \frac{L}{2} \, ,$$

$$\left[ \frac{2\alpha}{1 - \alpha(1 - \beta)} \right]^{\frac{L}{2}} = \prod_{b=1}^{\frac{L}{2}} (u_b + \delta) \, . \tag{15}$$

In terms of the solutions of the Bethe equations, the steady state has $E = 1$ and corresponds to [30]

$$z_a \longrightarrow 1 \, , \quad a = 1, \ldots, N. \tag{16}$$

## 2.3   Excited states for $L = 2N$ in the zero-momentum sector

We determine excited states by means of the following procedure. As is usually the case in Bethe ansatz solvable models [32–35], the roots characterizing the eigenstates approach a limiting distribution in the thermodynamic limit. In most cases studied in the literature, the root distribution can be described in terms of (half-odd) integer numbers that follow a given pattern that can be easily followed when $L$ is increased at fixed particle number. In order for this to work it is essential to have a suitable parametrization of the Bethe equations such that, upon taking logarithms with suitably defined branch cuts, one obtains a one-to-one correspondence between eigenstates and (half-odd) integer numbers [33, 34, 36]. Given the unusual form of our Bethe equations, it is not clear how to achieve this for the lifted TASEP. We therefore proceed in an iterative way. Let us assume that we have constructed a family of eigenstates at fixed density $\frac{N}{L} = \frac{1}{2}$ up to a given system size. From the roots we can determine the corresponding sets $\{\mu_\ell\}$, $\{\delta_\ell\}$ and $\{\beta_\ell\}$.

1. Using an extrapolation algorithm we determine $\mu_{L'}$, $\delta_{L'}$, $\beta_{L'}$ with (even) $L' \geq L + 2$ from the sequences $\{\mu_\ell\}$, $\{\delta_\ell\}$ and $\{\beta_\ell\}$.

2. We then determine the $L'$ roots of the polynomial equation (15). Out of these we select the $L'/2$ roots that most closely follow the pattern set out by the solution $\{u_1, \ldots, u_{L/2}\}$ for system size $L$. This gives us a starting point for solving the Bethe equations for system size $L'$.

3. We numerically solve the Bethe equations (15) for system size $L'$.

The starting point for our procedure is established by numerically solving the Bethe equations for small system sizes $L = 10, 12, 14$ and identifying eigenstates that belong to the same family "by hand".

For $\alpha \neq 1/2$ the eigenvalue obtained for large sizes $L$ is then fitted to the functional form

$$E(L) = c_1 L^{-5/2} + c_2 L^{-3} + c_3 L^{-7/2} \, . \tag{17}$$

In order to ensure that we are "following" the correct state by our iterative procedure, we determine a set of integers obtained by taking the logarithm of the Bethe equations

$$2\pi i I_j = \frac{L}{2} \ln \left( u_j - 1 \right) + \frac{L}{2} \ln \left( -u_j - 1 \right) - \ln(\mu) - \ln \left( u_j + \delta \right) , \quad j = 1, \ldots, \frac{L}{2} \, . \tag{18}$$

In practice, we choose a variety of different branch cuts, which gives rise to different definitions of the $I_j$. Importantly, the definition (18) does not rule out that the same integer $I_j$ occurs more than once, but this is not relevant for our purposes. In the following, we focus on three classes of excited states:

- **State 1:**

  The first state has zero momentum and

  $$E, \mu, \delta, \beta \in \mathbb{R} \,. \tag{19}$$

  For $L = 4n + 2$ with $n$ a positive integer, it is associated with a sequence of $I_j$ (18) of the form

  $$-\frac{L-2}{4}, -\frac{L-2}{4} + 1, \ldots, -2, 0, 0, 0, 2, 3 \ldots, \frac{L-2}{4} \,. \tag{20}$$

  We have considered this state for $\alpha \leq \frac{1}{2}$.

- **State 2:**

  The second state has zero momentum and

  $$E, \mu, \delta, \beta \in \mathbb{R} \,. \tag{21}$$

  For $L = 4n + 2$ with $n$ a positive integer, it is associated with a sequence of $I_j$ of Eq. (18) with the form

  $$-\frac{L-2}{4}, -\frac{L-2}{4} + 1, \ldots, \frac{L-2}{4} \,. \tag{22}$$

  We have considered this state for $\alpha > \frac{1}{2}$. We note that results for the same state at $\alpha = 0.9$ were already reported in Ref. [30].

- **State 3:**

  The third state has momentum $P = \frac{2\pi}{L}$ [1] and

  $$E, \mu, \delta, \beta \in \mathbb{C} \,. \tag{23}$$

  For $L = 4n + 2$ with $n$ a positive integer, it is associated with a sequence of $I_j$ of Eq. (18) of the form

  $$-\frac{L-2}{4}, -\frac{L-2}{4} + 1, \ldots, \frac{L-2}{4} \,. \tag{24}$$

  This state is of particular interest at $\alpha = 1/2$.

### 2.3.1  State 1 at $\alpha = 0.1$

In Fig. 2, we show results for State 1 for system sizes $L \leq 172$. As noted above, the imaginary part of the eigenvalue vanishes, while the $L$-dependence of the real part of $\ln(E)$ is given by Eq. (17) with fit parameters

$$c_1 = 6.07924 \,, \quad c_2 - 1.7243 \,, \quad c_3 = -25.554 \,. \tag{25}$$

As shown in Fig. 2, this provides an excellent fit to the data.

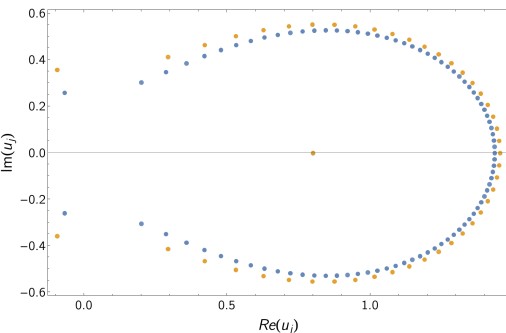 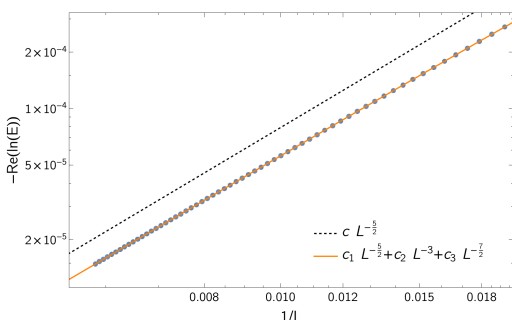

Figure 2: Sets of roots $\{u_j | 1 \leq j \leq L/2\}$ (left panel) and scaling of the eigenvalue of the transition matrix with system size (right panel) for the excited state of Eq. (20) for $L = 2N$ and $\alpha = 0.1$, with $L = 172$ (blue) and $L = 92$ (yellow). For large $L$, the roots approach a non-trivial contour in the complex plane. The orange line in the right panel is the fit of Eq. (25) to the functional form of Eq. (17).

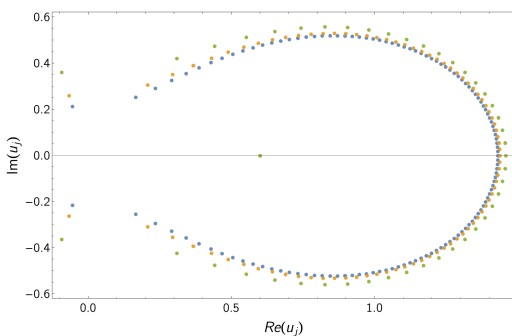 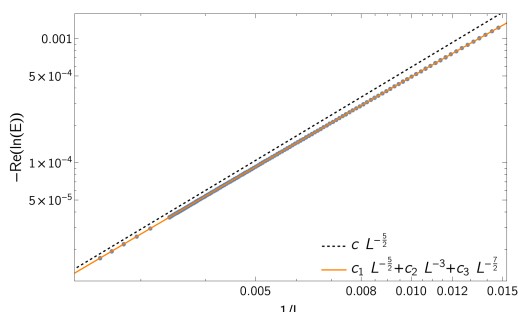

Figure 3: Sets of roots $\{u_j | 1 \leq j \leq L/2\}$ (left panel) and scaling of the eigenvalue of the transition matrix with system size (right panel) for the excited state of Eq. (20) for $L = 2N$ and $\alpha = 0.2$, with $L = 248$ (blue), $L = 172$ (yellow) and $L = 92$ (green). The orange line in the right panel is the fit of Eq. (26) to the functional form of Eq. (17).

### 2.3.2 State 1 at $\alpha = 0.2$

In Fig. 3, we show results for State 1 for system sizes $L \leq 380$. As noted above, the imaginary part of the eigenvalue vanishes, while the $L$-dependence of the real part of $\ln(E)$ is given by (17) with fit parameters

$$c_1 = 58.6837 , \quad c_2 = -51.4679 , \quad c_3 = -347.858 . \tag{26}$$

As can be seen in Fig. 3, this provides an excellent fit to the data. We observe a $L^{5/2}$ scaling of $\text{Re}\left[\ln(E)\right]$ for sufficiently large $L > L_{co}(\alpha)$. The crossover scale $L_{co}^{(1)}(0.2)$ is significantly larger than the one for $\alpha = 0.1$.

### 2.3.3 State 1 at $\alpha = 0.4$

In Fig. 4, we show results for State 1 for system sizes $L \leq 1000$. The imaginary part of the eigenvalue vanishes, while the $L$-dependence of the real part of $\ln(E)$ is given by (17) with fit parameters

$$c_1 = 920.557 , \quad c_2 = -8286.03 , \quad c_3 = 23384.5 . \tag{27}$$

---

[1]There is a corresponding state with $P = -\frac{2\pi}{L}$ that is related by complex conjugation of the roots and of $E$.

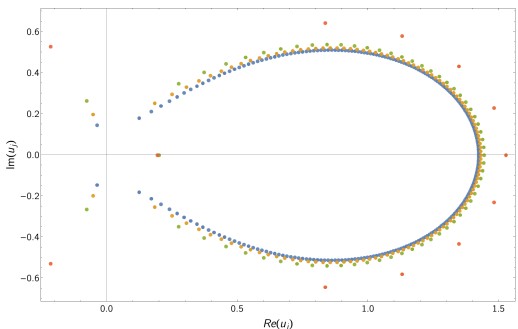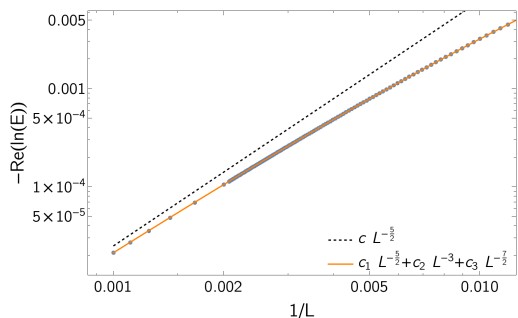

Figure 4: Sets of roots $\{u_j|1 \le j \le L/2\}$ (left panel) and scaling of the eigenvalue of the transition matrix with system size (right panel) for the excited state of Eq. (18) for $L = 2N$ and $\alpha = 0.4$, with $L = 484$ (blue), $L = 244$ (yellow), $L = 124$ (green) and $L = 24$ (red). For large $L$, the roots approach a non-trivial contour in the complex plane.

As can be seen in Fig. 4, this provides an excellent fit to the data, but the asymptotic $L^{-5/2}$ scaling is approached only for very large values of $L > L_{\text{co}}^{(1)}(0.4)$, where the crossover scale is much larger than for $\alpha = 0.2$ and for $\alpha = 0.1$. These observations are compatible with a crossover scale that diverges as $\alpha$ approaches the critical value $\alpha_{\text{crit}} = 1/2$:

$$\lim_{\alpha \to \frac{1}{2}} L_{\text{co}}^{(1)}(\alpha) = \infty. \tag{28}$$

### 2.3.4   State 1 at $\alpha = 0.5$

In Fig. 5, we show results for State 1 for system sizes $L \le 1000$. The imaginary part of the

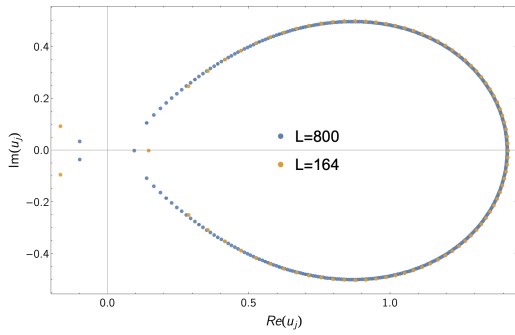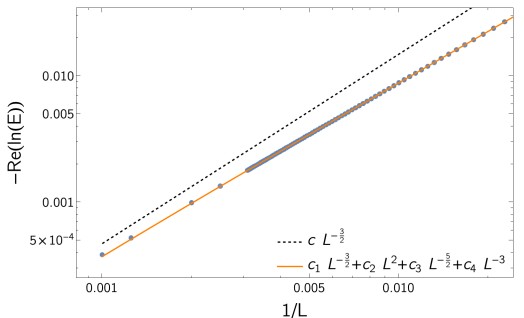

Figure 5: Sets of roots $\{u_j|1 \le j \le L/2\}$ (left panel) and scaling of the eigenvalue of the transition matrix with system size (right panel) for the excited state of Eq. (20) for $L = 2N$ and $\alpha = 0.5$. The orange line in the right panel is the fit of Eq. (30) to the functional form of Eq. (29).

eigenvalue is zero, while the $L$-dependence of the real part of $\ln(E)$ is fitted to the functional form

$$E(L) = c_1 L^{-3/2} + c_2 L^{-2} + c_3 L^{-5/2} + c_4 L^{-3} , \tag{29}$$

where

$$c_1 = 14.3328 , \quad c_2 = -93.2641 , \quad c_3 = 493.579 , \quad c_4 = -1052.21 . \tag{30}$$

As can be seen in Fig. 5, this provides an excellent fit to the data. Our numerical results are compatible with an asymptotic $L^{-3/2}$ scaling.

### 2.3.5 State 3 at $\alpha = 0.5$

In Fig. 6 we show results for State 3 for system sizes $L \leq 800$. The imaginary part of the

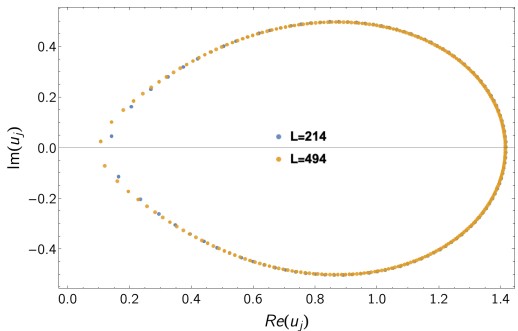
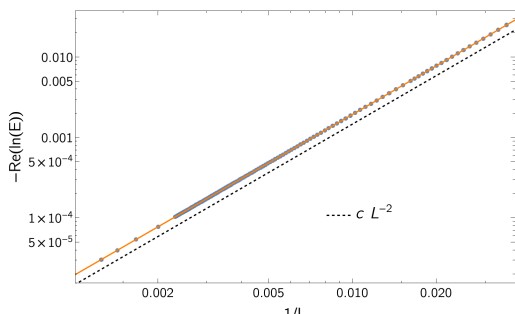

Figure 6: Sets of roots $\{u_j | 1 \leq j \leq L/2\}$ (left panel) and scaling of the eigenvalue of the transition matrix with system size (right panel) for the excited state of Eq. (24) for $L = 2N$ and $\alpha = 0.5$. The orange line in the right panel is the fit of Eq. (32) to the functional form of Eq. (31).

eigenvalue vanishes, while the $L$-dependence of the real part of $\ln(E)$ is fitted to the functional form

$$E(L) = c_1 L^{-2} + c_2 L^{-5/2} + c_3 L^{-3} , \tag{31}$$

where

$$c_1 = 19.9061 , \quad c_2 = -3.49057 , \quad c_3 = 20.97 . \tag{32}$$

As can be seen in Fig. 6, this provides an excellent fit to the data.

### 2.3.6 State 2 at $\alpha = 0.7$

In Fig. 7 we show results for State 2 for system sizes $L \leq 1000$. The imaginary part of the

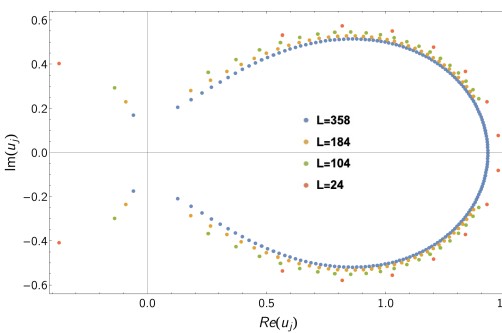
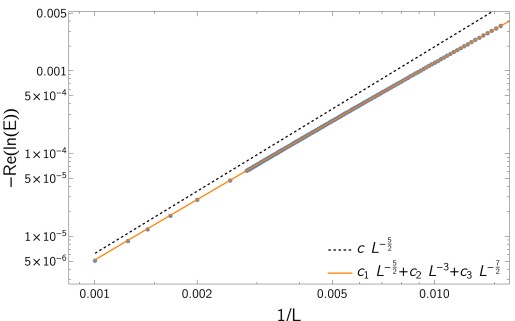

Figure 7: Sets of roots $\{u_j | 1 \leq j \leq L/2\}$ (left panel) and scaling of the eigenvalue of the transition matrix with system size (right panel) for the excited state of Eq. (18) for $L = 2N$ and $\alpha = 0.7$. The orange line in the right panel is the fit of Eq. (33) to the functional form of Eq. (17).

eigenvalue vanishes, while the $L$-dependence of the real part of $\ln(E)$ is given by Eq. (17) with fit parameters

$$c_1 = 189.245 , \quad c_2 = -750.725 , \quad c_3 = 1353.99 . \tag{33}$$

As can be seen in Fig. 7, this provides an excellent fit to the data, but the asymptotic $L^{-5/2}$ scaling is approached only for large values of $L > L_{co}^{(2)}(0.7)$, where the crossover scale is much larger than for $\alpha = 0.8$.

### 2.3.7 $\alpha = 0.8$

In Fig. 8, we show results for State 2 for system sizes $L \leq 800$. The imaginary part of the

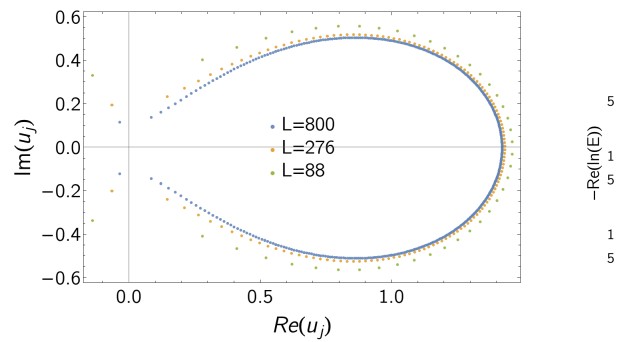 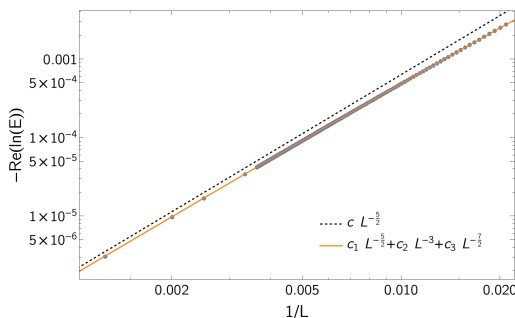

Figure 8: Sets of roots $\{u_j | 1 \leq j \leq L/2\}$ (left panel) and scaling of the eigenvalue of the transition matrix with system size (right panel) for the excited state of Eq. (18) for $L = 2N$ and $\alpha = 0.8$. The roots are shown for $L = 800$ (blue), $L = 276$ (yellow) and $L = 88$ (green).

eigenvalue vanishes, while the $L$-dependence of the real part of $\ln(E)$ is given by Eq. (17) with fit parameters

$$c_1 = 61.2629 , \quad c_2 = -122.315 , \quad c_3 = 57.1236 . \tag{34}$$

As can be seen in Fig. 8, this provides an excellent fit to the data.

### 2.3.8 Excitation gap

The above analysis provides us with bounds for the spectral gap

$$\Delta = \mathrm{Re}\ln[E^*] , \tag{35}$$

where $E^* \neq 1$ is the eigenvalue of the transition matrix with magnitude closest to one. We conclude that, for large $L$, we must have

$$\Delta \leq \begin{cases} \mathrm{const}\, L^{-5/2} & \text{if } \alpha \neq \frac{1}{2} , \\ \mathrm{const}\, L^{-2} & \text{if } \alpha = \frac{1}{2} . \end{cases} \tag{36}$$

## 3 Resolving the discrepancy between Bethe Ansatz and MC simulations

The bound for the spectral gap of the transition matrix obtained from the Bethe ansatz solution

$$\Delta\left(\alpha = \frac{1}{2}\right) \leq \mathrm{const}\, L^{-2} , \tag{37}$$

differs from the autocorrelation time of the structure factor and similar observables, which, according to our MC simulations, behave as

$$\tau_{\mathrm{IAC}} \sim L^{3/2} . \tag{38}$$

A possible explanation is that the eigenvector(s) that corresponds to eigenvalues such that $\mathrm{Re}\ln(E) \propto L^{-2}$ have negligible overlap with the large-scale density modes tracked by the structure factor, and there are too few of them.

## 3.1   Spectral representation of linear-response functions

In order to proceed, it is useful to view the configurations $\{\vec{j}, n\}$ as orthonormal basis vectors in a linear vector space

$$|\vec{j}; n\rangle , \qquad \langle \vec{k}; m | \vec{j}; n\rangle = \delta_{n,m} \delta_{\vec{k}, \vec{j}} . \tag{39}$$

We then define the uniform state by

$$|u\rangle = \sum_{\vec{j}, n} |\vec{j}; n\rangle . \tag{40}$$

The normalized steady state for $L = 2N$ is

$$|P_{\mathrm{SS}}\rangle = \frac{1}{\mathcal{N}} |u\rangle , \quad \mathcal{N} = \frac{L}{2} \binom{L}{L/2} . \tag{41}$$

The master equation then reads

$$\frac{d}{dt} |P(t)\rangle = \hat{T} |P(t)\rangle . \tag{42}$$

The observables of interest act on states as

$$\hat{\mathcal{A}} |\vec{j}; n\rangle = v_{\mathcal{A}}(\vec{j}, n) |\vec{j}; n\rangle . \tag{43}$$

Dynamical susceptibilities in equilibrium can then be written in the form

$$\chi_{\mathcal{A}\mathcal{B}}(t) = \langle u | \hat{\mathcal{A}} \hat{T}^t \hat{\mathcal{B}} | P_{\mathrm{SS}}\rangle . \tag{44}$$

A particular example considered in Ref. [30] and discussed above is

$$\mathcal{A} = \mathcal{B} = \hat{S}(-Q)\hat{S}(Q) , \quad \hat{S}(Q) |\vec{j}; n\rangle = \frac{1}{\sqrt{L}} \sum_{n=1}^{N} e^{iQ j_n} |\vec{j}; n\rangle . \tag{45}$$

It is useful to express these in a spectral representation in terms of eigenstates of the transition matrix $\hat{T}$. Assuming that $\hat{T}$ is diagonalizable by a similarity transformation (which we observe to hold for small $L$), we can take

$$\hat{T} |R_n\rangle = E_n |R_n\rangle , \quad \langle L_n | T = E_n \langle L_n | , \quad \langle L_n | R_m\rangle = \delta_{n,m} , \quad \mathbb{1} = \sum_n |R_n\rangle \langle L_n | . \tag{46}$$

This leads to the following spectral representation of the dynamical susceptibility

$$\chi_{\mathcal{A}\mathcal{B}}(t) = \sum_n \langle u | \hat{\mathcal{A}} | R_n\rangle \langle L_n | \hat{\mathcal{B}} | P_{\mathrm{SS}}\rangle \, e^{t \ln(E_n)} . \tag{47}$$

In practice we will normalize the right eigenstates

$$\langle R_n | R_n\rangle = 1 , \tag{48}$$

and the normalization of the $\langle L_n |$ then follows from (46). Our analysis of the Bethe ansatz equations suggests that the spectral gap scales as a power-law in $L$

$$\Delta \propto L^{-\alpha} . \tag{49}$$

Hence, in order to measure the correlation time $\tau_{\mathrm{corr}} \sim \Delta^{-1}$ in Monte Carlo simulation we need to consider times such that

$$t > L^{\alpha} . \tag{50}$$

As we are dealing with an interacting many-particle system, typical matrix elements are expected to scale as

$$\langle u|\hat{\mathcal{A}}|R_n\rangle\langle L_n|\hat{\mathcal{B}}|P_{SS}\rangle = \mathcal{O}\big(e^{-\gamma L}\big)\,. \tag{51}$$

As exponentially small (in $L$) effects are not accessible in Monte Carlo simulations on the large systems we are interested in, the $L$-dependence of the correlation time $\tau_{\mathrm{corr}}$ can be measured only if one of the following conditions is met:

1. There exists an exponentially large number of eigenstates $|R_n\rangle$ whose eigenvalues $E_n$ have the same scaling as the spectral gap

$$\mathrm{Re}\big(\ln(E_n)\big) \propto L^{-\alpha}\,. \tag{52}$$

   Summing over them in the spectral representation then gives a $\mathcal{O}(L^0)$ contribution to the susceptibility that at behaves as

$$\chi_{\mathcal{A}\mathcal{B}}(t) \sim c_1 e^{-c_2 t/L^\alpha}\,, \quad t > L^\alpha\,. \tag{53}$$

2. The number of eigenstates that have the same scaling as the spectral gap scales as a power of the system size, but they give rise to atypically large matrix elements

$$\langle u|\hat{\mathcal{A}}|R_n\rangle\langle L_n|\hat{\mathcal{B}}|P_{SS}\rangle = \mathcal{O}\big(L^{-\beta}\big)\,. \tag{54}$$

   In this case summing over these eigenstates again gives a contribution to the susceptibility of the form of Eq. (53), which dominates the late-time behaviour.

We now will present some results that support neither scenario in the lifted TASEP at $\alpha = \alpha_{\mathrm{crit}}$. This suggests that the relaxation time (the inverse gap) is not visible in the decay of dynamical susceptibilities and related observables extracted from Monte Carlo simulations.

## 3.2 Translational invariance and momentum eigenbasis

The translation operator $\hat{\tau}$ acts on basis states as

$$\hat{\tau}|\vec{j};n\rangle = \begin{cases} |j_1+1,\ldots,j_N+1;n\rangle & \text{if } j_N < L\,, \\ |1,j_1+1,\ldots,j_{N-1}+1;n+1\rangle & \text{if } j_N = L\,. \end{cases} \tag{55}$$

Here, the pointer variable $n+1$ is to be understood mod $L/2$. As we are working on an $L$-site lattice with periodic boundary conditions, we have $\hat{\tau}^L = \mathbb{1}$. A basis of momentum eigenstates is then given by

$$|q;\vec{j}_0\rangle = \frac{1}{\sqrt{L}}\sum_{k=0}^{L-1} e^{iqk}\hat{\tau}^k|\vec{j}_0;1\rangle\,, \quad q = \frac{2\pi m}{L}\,, \quad -\frac{L}{2} < m \le \frac{L}{2}\,, \tag{56}$$

$$\hat{\tau}|q;\vec{j}_0\rangle = e^{-iq}|q;\vec{j}_0\rangle\,, \tag{57}$$

where $\{\vec{j}_0;1\}$ is the set of configurations with the pointer particle located on site 1. This basis is useful because the Bethe ansatz states are by construction momentum eigenstates and we can gain insight into their structure by looking at them in the $\{|q;\vec{j}_0\rangle\}$ basis.

### 3.2.1    Action of observables on momentum eigenstates

The dynamical structure factor (45) is translationally invariant

$$\hat{\tau}\hat{S}(-Q)\hat{S}(Q)|q;\vec{j}_0\rangle = e^{-iq}\hat{S}(-Q)\hat{S}(Q)|q;\vec{j}_0\rangle\ , \tag{58}$$

which implies that

$$\langle u|\hat{S}(-Q)\hat{S}(Q)|q;\vec{j}_0\rangle \propto \delta_{q,0}\ . \tag{59}$$

Hence the eigenstate exhibiting $L^{-2}$ scaling (State 3) considered above does not contribute in the spectral representation of $\chi_{\mathcal{A}\mathcal{A}}$ with $\mathcal{A} = \hat{S}(-Q)\hat{S}(Q)$. However, the observable $\hat{S}(Q)$ (as well as other observables we have considered in our Monte Carlo simulations) does have non-vanishing matrix elements between $\langle u|$ and finite-momentum states

$$\langle u|\hat{S}(Q)|q;\vec{j}_0\rangle \propto \delta_{Q,-q}\ . \tag{60}$$

## 3.3    Structure of State 3 for small $L = 2N$

In the $P = 2\pi/L$ momentum sector, there are a total of

$$\frac{1}{2}\binom{L}{L/2} \tag{61}$$

eigenstates $\langle L_n|$ of the transition matrix. The particular state $\langle L_1|$ belonging to the State 3 family of Bethe states corresponds to the following solution of the Bethe equations (for the left eigenvectors)

- $L = 2N = 10$, $\ln(E) = -0.213449 + 0.702038i$

  $u_1 = 0.427136 - 0.873870i$, $u_2 = 0.564221 + 0.680842i$, $u_3 = 1.33637 + 0.597608i$,
  $u_4 = 1.35093 - 0.646121i$, $u_5 = 1.66292 - 0.005111i$ . $\tag{62}$

- $L = 2N = 12$, $\ln(E) = -0.145070 + 0.573101i$

  $u_1 = 0.421706 - 0.736742i$, $u_2 = 0.451233 + 0.565759i$, $u_3 = 1.11218 + 0.639909i$,
  $u_4 = 1.18183 - 0.655295i$, $u_5 = 1.52658 + 0.28296i$, $u_6 = 1.55750 - 0.23499i$ .
  $\tag{63}$

- $L = 2N = 14$, $\ln(E) = -0.105168 + 0.484332i$

  $u_1 = 0.389443 + 0.481886i$, $u_2 = 0.408221 - 0.644572i$, $u_3 = 0.963278 + 0.632744i$,
  $u_4 = 1.05742 - 0.647857i$, $u_5 = 1.38015 + 0.42379i$, $u_6 = 1.43968 - 0.36951i$ ,
  $u_7 = 1.55081 + 0.03846i$ . $\tag{64}$

In order to gain insight into the structure of all eigenstates in the $P = 2\pi/L$ momentum sector, we have determined their expressions in both the configuration basis (39) and the momentum eigenbasis (57) by numerically computing the overlaps

$$\langle\vec{j};m|R_n\rangle\ , \qquad \langle\frac{2\pi}{L};\vec{j}_0|R_n\rangle\ . \tag{65}$$

We first consider the amplitudes in the configuration basis. The state (62) has a set of amplitudes shown in Fig. 9. We see that in the product basis $\{\langle\vec{j};m|\}$ the distribution of angles is roughly uniform, while the distribution of magnitudes is strongly peaked around 0.028. The other states have distributions of $\langle\vec{j};m|R_n\rangle$ that look qualitatively different. Examples are shown in Fig. 11.

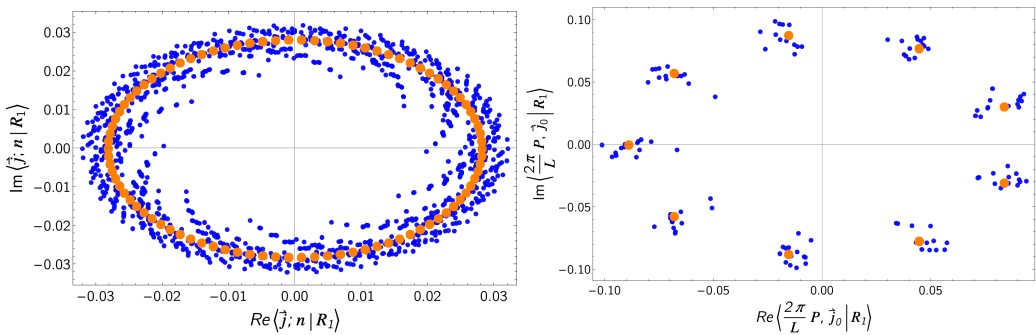

Figure 9: Plot of the real and imaginary parts of the amplitudes $\langle \vec{j}; m | R_1 \rangle$ (left panel, blue crosses) and $\langle \frac{2\pi}{L}; \vec{j}_0 | R_1 \rangle$ (right panel), where $|R_1\rangle$ is the eigenstate (62) ($L = 2N = 10$). The orange dots are, respectively, the amplitudes $\langle \vec{j}; m | F_{1,\psi} \rangle$ and $\langle \frac{2\pi}{L}; \vec{j}_0 | F_{1,\psi} \rangle$ for the variational state defined in Eq. (66) with $\psi = \pi/4$.

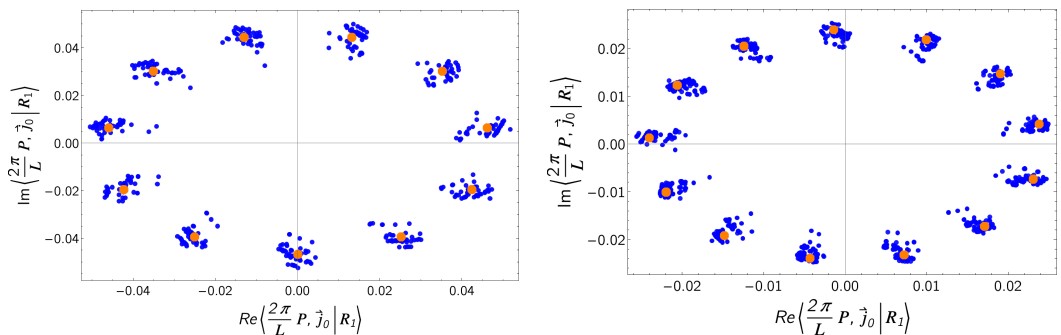

Figure 10: Real vs imaginary parts of the amplitudes $\langle \frac{2\pi}{L}; \vec{j}_0 | R_1 \rangle$ for $N = 6$ (left panel) and $N = 7$ (right panel), where $|R_1\rangle$ are the eigenstates (63) and (64). The orange dots are the amplitudes $\langle \frac{2\pi}{L}; \vec{j}_0 | F_{1,\psi} \rangle$ for $\psi = \frac{5\pi i}{22}$ and $\psi = \frac{3\pi i}{52}$ respectively.

### 3.3.1 Approximate structure of State 3

In order to understand the physical properties of State 3, we propose an approximate variational description of the form

$$|F_{r,\psi}\rangle = e^{i\psi} \sum_{\vec{j}_0} r_{\vec{j}_0} e^{i\phi_{\vec{j}_0}} |\frac{2\pi}{L}; \vec{j}_0\rangle , \tag{66}$$

where we choose

$$\phi_{\vec{j}_0} = \frac{4\pi X_{\vec{j}_0}}{L-1} , \quad X_{\vec{j}_0} = \sum_{m=2}^{L/2} \left( j_{0,m} - \frac{L+2}{2} \right) . \tag{67}$$

Here, $X_{\vec{j}_0}$ is obtained by removing the first site (pointer position) from the configuration $\{\vec{j}_0; 1\}$, and then summing over the distances of the remaining particles from the center of the lattice. In the configuration basis, the state reads

$$|F_{r,\psi}\rangle = e^{i\psi} \sum_{\{\vec{j};n\}} r_{\vec{j}_0} e^{i\phi_{\vec{j}_0} + \frac{2\pi i j_n}{L}} |\vec{j}; n\rangle , \quad \text{where } |\vec{j}; n\rangle = \hat{\tau}^{n-1} |\vec{j}_0; 1\rangle . \tag{68}$$

The amplitudes of the simplest variational state $|F_{1,\psi}\rangle$ are shown in Fig. 9 and Fig. 10 for $L = 2N = 10, 12, 14$, respectively, and are seen to approximately track those of State 3.

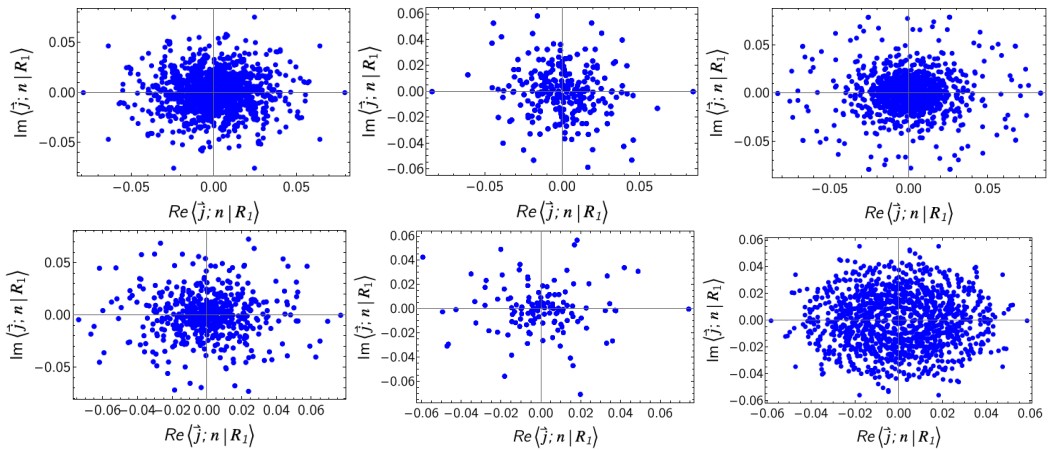

Figure 11: Plot of the real and imaginary parts of the amplitudes $\langle \vec{j}; m | R_n \rangle$ for randomly selected other right eigenstates $|R_n\rangle$.

### 3.3.2  Overlaps involving State 3

We now turn to the overlaps

$$\omega_N(Q) \equiv \langle u | \hat{S}(-Q) | R_1 \rangle \langle L_1 | \hat{S}(Q) | P_{SS} \rangle \, , \tag{69}$$

where

$$\hat{S}(Q) | \vec{j}, n \rangle = \frac{1}{\sqrt{L}} \sum_{k=1}^{N} e^{iQj_k} | \vec{j}, n \rangle \, . \tag{70}$$

By momentum conservation this vanishes unless $Q = 2\pi/L$. Numerically, we find for small systems

$$\omega_5 \left( \frac{2\pi}{L} \right) = 2.69504 \times 10^{-6} \, , \; \omega_6 \left( \frac{2\pi}{L} \right) = 1.71279 \times 10^{-7} \, , \; \omega_7 \left( \frac{2\pi}{L} \right) = 1.11663 \times 10^{-8}. \tag{71}$$

These are compatible with an exponential decay in $N$, and their smallness provides an explanation of why the gap associated with State 3 is not detected in MC simulations.

### 3.4  Dynamical response in the lifted TASEP

Our analysis is compatible with the following scenario, which is premised on the assumption that there is at most a polynomial number number of eigenstates of the transition matrix with eigenvalues such that $\mathrm{Re}\ln(E_n) \sim L^{-2}$, and their matrix elements are exponentially small in $L$:

$$\chi_{AB}(t) \sim \begin{cases} c_1 e^{-c_2 t/L^{3/2}} & \text{if } 1 \ll t \sim L^{3/2} \, , \\ c_3 e^{-\gamma L} e^{-c_4 t/L^2} & \text{if } L^{5/2} \lesssim t \, . \end{cases} \tag{72}$$

Here $c_j = \mathcal{O}(1)$. In this scenario the eventual exponential decay of the susceptibility with the relaxation time (inverse spectral gap) is not detectable by numerical methods because the susceptibility is too small to be reliably computed in the relevant time window.

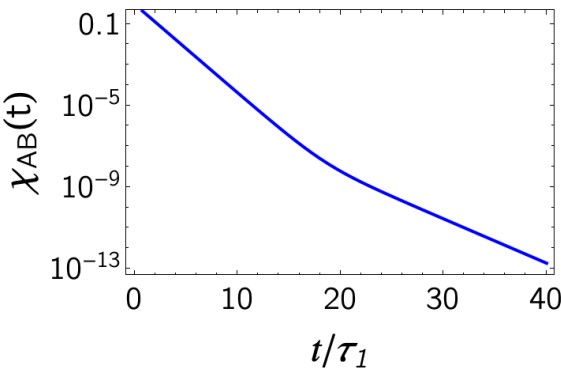

Figure 12: Cartoon of proposed behaviour of dynamical response functions $\chi_{\mathcal{AB}}(t)$ in equilibrium of the lifted TASEP. A exponential decay with rate $\tau_1^{-1} \sim L^{-3/2}$ over a large intermediate time window is followed by the true asymptotic decay with rate $\tau_2^{-1} \sim L^2$. The latter regime is undetectable numerically because the susceptibility is already extremely small.

## 4   Continuum limit of the lifted TASEP

In order to facilitate the continuum limit, it is useful to introduce a step-size $\Delta t$ for the time evolution and rewrite the master equation as

$$P(t + \Delta t) = TP(t) = P(t) + \Delta t \, \frac{T-1}{\Delta t} P(t) \,. \tag{73}$$

Rearranging then gives

$$\frac{P(t + \Delta t) - P(t)}{\Delta t} = MP(t) \,, \qquad M = \frac{T-1}{\Delta t}. \tag{74}$$

The eigenvectors of $M$ are the same as the ones of $T$, and the eigenvalues are related by

$$\mathcal{E} = \frac{E-1}{\Delta t} \,. \tag{75}$$

### 4.1   Taking the continuum limit

We introduce a lattice spacing $a_0$, such that the sites of our $L$-site lattice are at positions $ja_0$. The physical length of the lattice is then

$$\mathfrak{L} = La_0 \,. \tag{76}$$

The continuum limit corresponds to taking $a_0 \to 0$, $L \to \infty$, while keeping $\mathfrak{L}$ fixed. In the continuum limit it is useful to rewrite Eq. (9) for the amplitudes. Introducing continuous particle positions as

$$x_{j_n} = a_0 j_n \,, \quad n = 1, \dots, N \tag{77}$$

we have

$$\psi_a(\boldsymbol{x}) = \sum_{Q \in S_N} A_a(Q) \prod_{j=1}^{N} e^{\frac{\ln(z_j)}{a_0} x_{Q_j}} \,. \tag{78}$$

In order to retain a non-trivial dependence on the particle positions $x_j$ we introduce rescaled rapidity variables $u_j$ by

$$\ln(z_j) = u_j a_0 \,. \tag{79}$$

This results in

$$\prod_{j=1}^{N}\left[E-\frac{1-\alpha}{z_j}\right]=\prod_{j=1}^{N}\left(E-(1-\alpha)+a_0(1-\alpha)u_j\right),$$

$$\prod_{k=1}^{N}\frac{\alpha}{z_k}=\alpha^N e^{-a_0\sum_{k=1}^{N}u_k}=\alpha^N\left[1-a_0\sum_{k=1}^{N}u_k+\mathcal{O}(a_0^2)\right],\tag{80}$$

which in turn suggests the expansion

$$E=1+a_0\epsilon.\tag{81}$$

In order to obtain a non-trivial scaling limit we recall that the eigenvalues of $M$ are related to $E$ by

$$\mathcal{E}=\frac{E-1}{\Delta t}=\frac{a_0}{\Delta t}\epsilon.\tag{82}$$

This means that the continuum limit requires going over to a continuous-time description

$$\Delta t, a_0\to 0\,,\qquad\frac{a_0}{\Delta t}=v=\text{ fixed.}\tag{83}$$

Here $v$ is a characteristic propagation velocity. Finally, we rescale the pullback parameter $\alpha$ as

$$\alpha=a_0\mathfrak{a}.\tag{84}$$

The Bethe equations then become

$$e^{\mathfrak{L}u_j}=(-1)^{N+1}\left(1+\frac{\epsilon+u_j}{\mathfrak{a}}\right)\prod_{b=1}^{N}\frac{\epsilon+u_b}{\epsilon+u_j}\,,$$

$$1=\prod_{j=1}^{N}\left(1+\frac{\epsilon+u_j}{\mathfrak{a}}\right).\tag{85}$$

We may change variables to bring the Bethe equations into a nicer form

$$w_j=\frac{\epsilon+u_j}{\mathfrak{a}}.\tag{86}$$

Then

$$e^{\mathfrak{L}(\mathfrak{a}w_a-\epsilon)}=(-1)^{N+1}(1+w_a)\prod_{b=1}^{N}\frac{w_b}{w_a}\,,$$

$$1=\prod_{j=1}^{N}(1+w_j).\tag{87}$$

### 4.1.1 One and two-particle sectors

For $N=1$ we have $w_1=0$, which translates into

$$\epsilon=u_1=\frac{2\pi in}{\mathfrak{L}}\,,\quad n\in\mathbb{Z}.\tag{88}$$

In the two-particle sector $N=2$ the Bethe equations can be reduced to a simple quadratic equation

$$\epsilon=\mathfrak{a}\frac{w_1+w_2}{2}\bmod\frac{2\pi i}{\mathfrak{L}}\,,\qquad w_2=-\frac{w_1}{1+w_1}\,,\qquad w_1^2+2w_1\left(1+\frac{\pi in}{\mathfrak{a}\mathfrak{L}}\right)+\frac{2\pi in}{\mathfrak{a}\mathfrak{L}}=0.\tag{89}$$

## 4.2   Master equation in the continuum limit

In the continuum limit configurations are labelled by $N$ co-ordinates $x_j$ and the position $a$ of the pointer among the particles. Let us denote the probability distribution function corresponding to a configuration $(\boldsymbol{x}, a)$ by

$$P_a(\boldsymbol{x}) \, . \tag{90}$$

By taking the "naive" continuum limit of the discrete master equation we obtain

$$\frac{1}{v} \frac{\partial P_a(\boldsymbol{x}; t)}{\partial t} = -\frac{\partial}{\partial x_a} P_a(\boldsymbol{x}; t) + \mathfrak{a}[P_{a+1}(\boldsymbol{x}; t) - P_a(\boldsymbol{x}; t)] + \delta(x_a - x_{a-1})\big[P_{a-1}(\boldsymbol{x}; t) - P_a(\boldsymbol{x}; t)\big] \, . \tag{91}$$

Some of the relevant steps are

1. On the lattice the probability of a configuration $(\boldsymbol{j}, a)$ is $\psi_a(\boldsymbol{j})$. In the continuum limit the particle positions are given by

$$x_n = j_n a_0 \, . \tag{92}$$

2. The discrete time master equation on the lattice is

$$\psi_a(\boldsymbol{j}; t + \Delta t) = \bar{\alpha} \psi_{a-1}(\boldsymbol{j}; t) \, \delta_{j_a, j_{a-1}+1} + \bar{\alpha} \psi_a(\dots, j_a - 1, \dots; t) \, (1 - \delta_{j_a, j_{a-1}+1})$$
$$+ \alpha \psi_{a+1}(\dots, j_{a+1} - 1, \dots; t) \, (1 - \delta_{j_{a+1}, j_a+1}) + \alpha \psi_a(\boldsymbol{j}; t) \, \delta_{j_{a+1}, j_a+1} \, . \tag{93}$$

Setting

$$P_a(\boldsymbol{x}; t) = \psi_a(\boldsymbol{j}; t) \, , \tag{94}$$

and using that (this identity is understood in terms of a summation/integration over a test function)

$$\delta_{j,k} \to a_0 \delta(x - x') \tag{95}$$

we obtain

$$P_a(\boldsymbol{x}; t + \Delta t) = (1 - a_0 \mathfrak{a}) P_{a-1}(\boldsymbol{x}; t) \, a_0 \delta(x_a - x_{a-1} - a_0)$$
$$+ (1 - a_0 \mathfrak{a}) P_a(\dots x_a - a_0, \dots; t) \, \big(1 - a_0 \delta(x_a - x_{a-1} - a_0)\big)$$
$$+ \mathfrak{a} P_{a+1}(\dots, x_{a+1} - a_0, \dots; t) \, \big(1 - a_0 \delta(x_{a+1} - x_a - a_0)\big)$$
$$+ \mathfrak{a} P_a(\boldsymbol{x}; t) \, a_0 \delta(x_{a+1} - x_a - a_0) \, . \tag{96}$$

3. Finally, we expand in $a_0$ and drop all terms of order $a_0^2$ to arrive at Eq. (91).

## 4.3   Markov-process interpretation of the continuum limit

As introduced in Section 4.1, we consider the continuum limit of the lifted TASEP of $N$ particles on $L$ lattice sites, where $L \to \infty$. In this limit, it samples the partition function of a gas of hard spheres with vanishing radius $\sigma \to 0$. In the original model with $L$ sites, and in terms of the original scales of length and time (see Ref. [30, Sec. III.A]), the mean activiy drift per single step is given by

$$\langle v_\rightarrow \rangle = -\alpha \frac{L}{N} + 1 \, . \tag{97}$$

Introducing the lattice constant $a_0$ and the continuous time with a velocity $v = 1$ as in Eqs (76) and (83), this yields

$$\langle v_\rightarrow \rangle = -\mathfrak{a} \frac{\mathfrak{L}}{N} + 1 \, . \tag{98}$$

In the continuum limit, the lifted TASEP becomes equivalent to the factor-field event-chain Monte Carlo algorithm for zero-diameter one-dimensional hard spheres [11], and the Markov chain governing the lifted TASEP turns into a continuous-time Markov process. The active sphere moves with positive velocity $v = 1$ either until it collides with its right-hand neighbour or until a Poisson clock of intensity $\mathfrak{a}$ moves the pointer to its left-hand neighbour. The collision between zero-diameter objects is of course somewhat artificial as, at the collision, the particle coordinates are the same.

For event-chain Monte Carlo, the pointer drift is an estimator for the system pressure [13] and indeed, for $\mathfrak{a} = 0$, we recover the well-known formula

$$\beta P = \frac{\partial}{\partial \mathfrak{L}} Z(N, \mathfrak{L}, \sigma) = \frac{N}{\mathfrak{L}} \langle v_\rightarrow \rangle = \frac{N}{\mathfrak{L}}, \tag{99}$$

where the hard-disk partition function is $Z(N, \mathcal{L}) = \mathcal{L}^N$, in other words the partition function of one-dimensional hard spheres (for $\sigma = 0$) first computed by Tonks in 1936 [37, 38].

For finite $\mathfrak{a}$, the partition function is of a modified hard-sphere model

$$U = \sum_{k,l} U^{\text{hs}}(x_k, x_l) + \mathfrak{a} \sum_{k=1}^{N} (x_k - x_{k-1}), \quad x_{N+1} \equiv x_1 + \mathfrak{L}. \tag{100}$$

Here, in addition to the hard-sphere potential $U^{\text{hs}}$, there are linear attractions of strength $\mathfrak{a}$ between nearest neighbours called factor fields, which do not modify the particle statistics, as they sum up to $\mathfrak{aL}$ (see Ref. [39] for a discussion). Even for the motion of a single particle $x_k$, the factor-field term in Eq. (100) remains unchanged. Nevertheless, it influences the lifted-TASEP dynamics, in which four factors independently influence the motion, namely the factor field of $x_k$ with its forward neighbour (positioned at $x_{k+1}$, up to boundary conditions), the backward neighbour (at $x_{k-1}$), and identically for the hard-sphere interaction (see Section 5.1 for a discussion of this point in the language of the factorized Metropolis algorithm). The weight of each configuration is now

$$\pi(\{x_1, \ldots, x_N\}) = \exp(-\beta \mathfrak{a} \mathfrak{L}). \tag{101}$$

The partition function is:

$$Z_\mathfrak{a}(N, \mathcal{L}) = \mathcal{L}^N e^{-\beta \mathfrak{L} \mathfrak{a}}, \tag{102}$$

with the pressure (for $\beta = 1$):

$$P_\mathfrak{a}(N, \mathfrak{L}) = \partial \log [Z_\mathfrak{a}(N, \mathfrak{L})] / \partial \mathfrak{L}|_{\beta=1} = N/\mathcal{L} - \mathfrak{a}. \tag{103}$$

Multiplied with $\mathfrak{L}/N$, the pressure trivially equals the pointer drift, a finding that holds much beyond the trivial $\sigma = 0$ case treated here, and that remains valid in more than one dimension [13, 40].

The $\mathcal{O}(N^{3/2})$ scaling for the autocorrelation times of the structure factor has been observed in the hard-sphere model also at finite $\sigma$ at the critical factor field, which becomes, for large $N$, $\mathfrak{a} = N/(\mathfrak{L} - N\sigma)$ [11]. In many other one-dimensional models, such as the Lennard-Jones model or the harmonic chain [39], this optimal scaling was found for a critical factor field which again corresponds to a vanishing pointer drift and, thus, to vanishing pressure.

## 5  Generalized Lifted TASEP

In the present section, we generalize the lifted TASEP beyond hard-sphere interactions, but retain that, at each time step, a uniquely defined active particle moves in the forward direction.

In the continuum, this generalization carries over to arbitrary continuous pair interactions but also to many-body potentials. It is the factorized Metropolis algorithm [13]. On the lattice, only monotonous interactions between neighbouring particles allow for a generalization of the lifted TASEP. For these cases, we can set up a lifted algorithm which conserves the Boltzmann distribution as a stationary distribution.

## 5.1   Factorized Metropolis filter

In the GL-TASEP, configurations do not all have the same statistical weight, but rather Boltzmann weights given by the nearest-neighbour interactions that can be written as a potential:

$$\exp\left(-\beta U_{...,i,j,k,...}\right) = \pi_{...,i,j,k,...} = \cdots \pi_{j-i}\pi_{k-j}\cdots . \tag{104}$$

Here, $\ldots, i, j, k, \ldots$ are particle positions and the nearest-neighbour pairs, such as $(i, j)$ and $(j, k)$, are referred to as "factors". Periodic boundary conditions are understood. In the factorized Metropolis filter [13], each factor accepts the move individually with its Metropolis filter. A move of the active particle $j \to j + 1$ is thus accepted with probability

$$p^{\text{fact}}(j \to j + 1) = \min\left(1, \frac{\pi_{j+1-i}}{\pi_{j-i}}\right) \min\left[1, \frac{\pi_{k-(j+1)}}{\pi_{k-j}}\right] . \tag{105}$$

We have to avoid that both factors in Eq. (105) reject the move simultaneously, in which case the pointer cannot be re-attributed. We thus require the interaction to be monotonous and, for concreteness, repulsive:

$$\pi_k \geq \pi_l \quad \text{for } k > l. \tag{106}$$

In the factorized Metropolis filter of Eq. (105), the factor $(i, j)$ always accepts the (forward) move of particle $j$, so that Eq. (105) further simplifies to:

$$p^{\text{fact}}(j \to j + 1) = \min\left[1, \frac{\pi_{k-(j+1)}}{\pi_{k-j}}\right] = \frac{\pi_{k-(j+1)}}{\pi_{k-j}} =: p_{k-(j+1)} . \tag{107}$$

The lifted TASEP of Eqs (1) and (2) corresponds to the choice $\{\pi_0, \pi_1, \pi_2 \ldots\} = \{0, 1, 1, \ldots\}$

The move from forward distance 4 toward 3 is accepted with probability $p_3$, toward 2 with probability $p_2$, and toward 0 with probability $p_0$. In the hard-sphere case, $p_0 = 0$. The simplest generalization is when $0 < p_1 < 1$ and $p_k = 1$ for $k > 1$. In this case, we call $p := p_1$.

### 5.1.1   Definition of the GL-TASEP

As in the TASEP, we consider the evolution probabilities for a lifted configuration

$$\tag{108}$$

Again, a single particle $j$ is active (carries the pointer), but its advance can be rejected at a distance by its forward neighbour $k$. The backward neighbour does not reject the move under the factorized Metropolis algorithm, as the potential is supposed repulsive. In the first half-step of the move, the displacement of the active particle from $j$ to $j + 1$ is accepted with the Metropolis ratio of the weights $\pi_{k-(j+1)}/\pi_{k-j}$ (see Eq. (107)). If the move is rejected, the particle at $k$ obtains the pointer. The second half-step of the move performs a pullback move with probability $\alpha$. In total:

$$\tag{109}$$

(Here, we set $\overline{\alpha} = 1 - \alpha$ and $\overline{p} = 1 - p$.) The system in Eq. (109) requires only the positions of the three particles $i, \overrightarrow{j}, k$ in order to define the evolution of an arbitrary large system with periodic boundary conditions, in which $\overrightarrow{j}$ is active. Equation (109) shows that the rows of the transition matrix sum up to one, in other words that $(1, \ldots, 1)$ is a right eigenvector of $P$ with eigenvalue 1. The right-eigenvector equation corresponding to Eq. (109) is:

$$E\psi^R(i, \overrightarrow{j}, k) = p_{k-(j+1)}\alpha\,\psi^R(\overrightarrow{i}, j+1, k) + p_{k-(j+1)}\overline{\alpha}\,\psi^R(i, \overrightarrow{j+1}, k)$$
$$+ \left[1 - p_{k-(j+1)}\right]\alpha\,\psi^R(i, \overrightarrow{j}, k) + \left[1 - p_{k-(j+1)}\right]\overline{\alpha}\,\psi^R(i, j+1, \overrightarrow{k}), \qquad (110)$$

where only one of the $p_k$ appears in all four terms. For any choice of the parameters $p_1, p_2, \ldots$ in the GL-TASEP, we have three sets of such closed equations for the right eigenvectors.

## 5.2  Global balance for the GL-TASEP

To establish the global-balance condition, we use that the factorized Metropolis algorithm, although it is used exclusively in a non-reversible setting, nevertheless satisfies the detailed-balance condition. The following example illustrates this for three particles $(i, \overrightarrow{j}, k)$.

$$\qquad (111)$$

The coefficients of the $E = 1$ eigenvector are the stationary probabilities, so that the flow into configuration $(i, \overrightarrow{j}, k)$ is given by

$$\pi_{i,j-1,k}\, p^{\text{fact}}\left(i, \overrightarrow{j-1}, k \rightarrow i, \overrightarrow{j}, k\right)\overline{\alpha} \qquad (112)$$
$$+ \pi_{i,j,k}\left[1 - p^{\text{fact}}\left(\overrightarrow{i}, j, k \rightarrow \overrightarrow{i+1}, j, k\right)\right]\overline{\alpha} \qquad (113)$$
$$+ \pi_{i,j,k-1}\, p^{\text{fact}}\left(i, j, \overrightarrow{k-1} \rightarrow i, j, \overrightarrow{k}\right)\alpha \qquad (114)$$
$$+ \pi_{i,j,k}\left[1 - p^{\text{fact}}\left(i, \overrightarrow{j}, k \rightarrow i, \overrightarrow{j+1}, k\right)\right]\alpha\,. \qquad (115)$$

The detailed-balance condition allows one to turn around Eqs (112) and (114), which gives

$$\pi_{i,j,k}\, p^{\text{fact}}\left((i, \overrightarrow{j}), k \rightarrow (i, \overrightarrow{j-1}), k\right)\overline{\alpha} \qquad (116)$$
$$+ \pi_{i,j,k}\left[1 - p^{\text{fact}}\left((\overrightarrow{i}, j), k \rightarrow (\overrightarrow{i+1}, j), k\right)\right]\overline{\alpha} \qquad (117)$$
$$+ \pi_{i,j,k}\, p^{\text{fact}}\left(i, (j, \overrightarrow{k}) \rightarrow i, (j, \overrightarrow{k-1})\right)\alpha \qquad (118)$$
$$+ \pi_{i,j,k}\left[1 - p^{\text{fact}}\left(i, (\overrightarrow{j}, k) \rightarrow i, (\overrightarrow{j+1}, k)\right)\right]\alpha = \pi_{i,j,k}\,. \qquad (119)$$

In Eqs (116) and (117), as indicated, the interval $(i, j)$ is shortened from either side (and the sum of these two terms gives $\pi_{i,j,k}\overline{\alpha}$, and likewise, in Eqs (118) and (119), the interval $(j, k)$ is shortened from either side (with the sum equal to $\pi_{i,j,k}\alpha$). It follows that the flow into $(i, \overrightarrow{j}, k)$ equals $\pi_{i,j,k}$. Global balance is satisfied, and the GL-TASEP has the stationary distribution $\pi$.

   To numerically test for the possible integrability, we checked for the criterion proposed in Ref. [41] and computed complex ratios of eigenvalues, both separated into momentum sectors and merged for all momentum sectors (see Fig. 13). The absence of structure in the histogram points towards integrability, but a more careful analysis for larger system sizes is required to have any confidence in this. An investigation of the two and three-particle sectors did not reveal any clear signs of integrability [42].

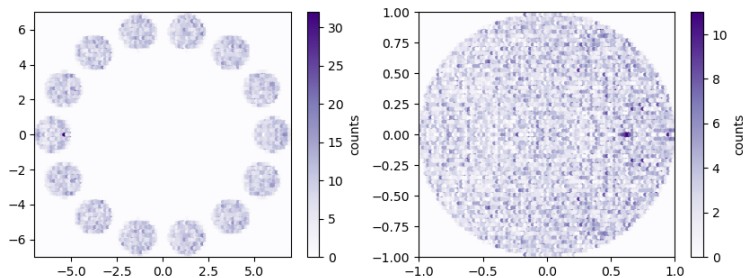

Figure 13: Complex eigenvalue-ratio statistics for the GL-Tasep. (Left): Eigenvalue ratios separated into momentum sectors. No structure is detected. (Right): Merged-momentum-sector eigenvalue ratios, which are again structureless. The parameters are $N = 7$, $L = 14$, for $\alpha = 0.2$ and $p = 0.8$.

## 6 Conclusions

In this work we have studied the lifted Tasep and its generalization, the GL-Tasep. In the lifted Tasep we have carried our a large-scale numerical analysis of the Bethe ansatz equations in order to show how the spectral gap $\Delta$ crosses over, as a function of $\alpha$, between the asymptotic $N^{-5/2}$ scaling of $\Delta$ at $\alpha \neq \alpha_{\text{crit}}$ and the $N^{-2}$ scaling at $\alpha = \alpha_{\text{crit}}$. An analogous crossover is observed in Monte Carlo simulations of the integrated autocorrelation time of the structure factor, but its scaling with particle number is seen to follow a $N^{3/2}$ law.

We have provided a possible explanation for this discrepancy by carefully analyzing, for small values of $N$, the properties of the eigenvector of the transition matrix that gives rise to the $N^{-2}$ scaling for large particle numbers at $\alpha = \alpha_{\text{crit}}$. We were able to "follow" this family of eigenvectors in $N$ by using the Bethe ansatz solution. We showed that for small particle numbers the contribution of the eigenvector of interest to dynamical susceptibilities is too small to be observed in Monte Carlo simulations. We propose that this smallness of the relevant matrix elements, combined with a small number of eigenvectors whose eigenvalues scale as $L^{-2}$, makes it essentially impossible to detect the asymptotic relaxation time numerically. To the best of our knowledge this is an unusual situation in stochastic processes of many interacting particles. However, such a scenario is not ruled out by any mathematical theorems, and may occur much more generally.

Another key result obtained in our work is the construction of an integrable continuum limit (in both space and time), which we related to the hard-sphere event-chain Monte Carlo algorithm. This allowed us to relate the pullback $\alpha$ is related to the pressure, and identify the critical pullback $\alpha_{\text{crit}}$ as corresponding to vanishing pressure, *cf.* [11].

Finally, we have generalized the lifted Tasep to a wide class of nearest-neighbour interactions, which leads to lifted Markov chains with non-trivial equilibrium steady states.

## Acknowledgements

This work was supported in part by the EPSRC under grant EP/X030881/1 (FHLE). FHLE thanks the Institut Henri Poincaré (UAR 839 CNRS-Sorbonne Université) and the LabEx CARMIN (ANR10-LABX-59-01) for their support. JG thanks the Rudolf Peierls Centre for Theoretical Physics for hospitality and financial support. Research of WK was supported by a grant from the Simons Foundation (Grant 839534, MET).

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
