# Peer review of "Lifted TASEP: long-time dynamics,generalizations, and continuum limit"

_SciPost Physics Core_

## Round 1 · Referee Report · Anonymous (Referee 1) · 2025-4-6

Report

The authors study in detail relaxation properties of an asymmetric exclusion process, called Lifted TASEP. In particular, they clarify convincingly an intriguing discrepancy between Bethe ansatz computations and Monte-Carlo simulations that were noted in earlier work by some of the same authors. This work will be interesting for the large scientific community working on particle systems far from thermal equilibrium.

Recommendation

Ask for minor revision

---

## Round 1 · Referee Report · Anonymous (Referee 2) · 2025-4-16

Report

This work is mainly a follow-up of the previous work in Ref [30]. The authors attempt to explain a previously noted disagreement between Monte Carlo simulations and Bethe-ansatz analysis regarding the asymptotic behavior of the relaxation time of the Lifted TASEP at half-filling for the critical pullback parameter \alpha_{crit}

The explanation relies on a numerically approximate examination of small-system examples and tries to show that the typical relevant eigenvectors have negligibly small overlap with common observables, making the ``true'' relaxation time essentially unmeasurable numerically.

While the subject is very interesting, I find the analysis presented here incomplete for the following reasons: 1. It's unclear why the 'breakdown' is unique to \alpha_{crit}. Please clarify what changes for other \alpha.

2.No practical proposal is provided for an alternative way to measure the relaxation time that would actually confirm the Bethe-ansatz analysis.

At the end of the paper, a generalization of the lifted TASEP is presented. However, the motivation behind this definition is not well laid out, and the results are very preliminary, which is not sufficient to fully appreciate the potential of this generalization. A simple question that could have been addressed is the following: Does this generalization correspond to a lifted version of any known model in the literature? Additionally, the abstract states, ``By tuning the pullback parameter in the GL-TASEP to a particular value we can again achieve a polynomial speedup in the time required to converge to the steady state.'' I did not find a follow-up to this statement in the main text. Please clarify this point.

Regarding the presentation of the article, I find that it can be improved for easier reading: a- The definition of the model at the top of page 2 is not clear enough: "a single particle being active. It carries a pointer which allows it to move in a forward direction or to undergo a collision. In the second part of the move, the pointer itself moves to the nearest neighbor to the left, which becomes the new active particle.'' This description does not fully match the figure, which can be confusing at first glance. In addition, It's note clearly stated that the model is built on a discrete-time TASEP, since the usual TASEP is in continuous time.

b-Figure 1 is a bit ambiguous. First, I assume the small horizontal dashes represent the simulation results; then what do the continuous lines represent? Are they just connecting the dashes? While one can accept that the values $\alpha = 0.34$ and $0.5$ indeed follow the predicted behavior, the other values do not have anything to compare.

c- Page 6 paragraphs are repetitive… condense or reorganize. Additionally, raw numerical data in the text do not offer particular insights, in particular on page 13.

More minor issues: - Equation (10) seems to me valid only for $N$ odd; otherwise, the solution $z_{\alpha} = 1, E = 1$ does not hold. Please clarify.

  • It is not clear where the fitting ansatz in Equation~(17) comes from.Is it purely empirical ?

  • There is a typo on page 12: ``the susceptibility that at best behaves as''

  • I am a bit confused by the notation: at the end of the paragraph ``Factorized Metropolis filter,'' it defines $p = p_1$. This suggests you do not need generic $p_k$ anymore, yet they are used in the following paragraph. Please clarify.

Requested changes

Address the comments in the report

Recommendation

Ask for minor revision

---

## Round 1 · Referee Report · Anonymous (Referee 3) · 2025-4-19

Strengths

Thorough numerical analyses of Bethe equations and related overlaps, and Monte Carlo simulations

Weaknesses

Lack of identification of a dynamical observable that scales in way clearly compatible with Bethe ansatz results at the critical value of the pullback.

Report

This manuscript provides a thorough analysis of a discrepancy that was observed in earlier work regarding the scaling of the spectral gap and the decay of dynamical susceptibilities. An proposed resolution is given in terms of very small overlaps with certain eigenstates, masking the true but almost unobservable asymptotic behaviour.

The authors provide substantial numerical evidence for the proposed resolution in the case of dynamical observables that they studied. I would have liked to see a dynamical observable at that does couple to the leading excitation at the critical value of the pullback. If it is difficult to identify such an observable, then some comments about this would be insightful.

The manuscript is certainly publishable but I am not entirely convinced from the presentation that the observed phenomenon is a significantly ubiquitous phenomenon and not an artefact of the particular model.

Requested changes

Please add an analysis of a dynamical observable at that does couple to the leading excitation at the critical value of the pullback, or a discussion if it is difficult to identify such an observable.

Recommendation

Publish (easily meets expectations and criteria for this Journal; among top 50%)

---

## Round 1 · Referee Report · Anonymous (Referee 4) · 2025-4-25

Weaknesses

1-The paper relies heavily on the previous work (Ref. [30])
by two of the authors.
2-The exposition of the results could be improven.

Report

The paper certainly merits publication. However, I recommend that the authors address some points to clarify the presentation and resolve a few inconsistencies.

Requested changes

\item The analogy with a quantum Hamiltonian is standard for the generator of continuous-time Markov chains. In the discrete-time case, however, the transition matrix corresponds to the exponential of minus the "Hamiltonian". This initially caused some confusion for me, as the spectral gap acts as an energy scale, and the eigenvalue of the transition matrix might be better denoted by a symbol like $\lambda$, rather than $E$, to avoid misleading associations.

\item In Eqs. (17), (29), and (31), I believe the left-hand side should be $-\text{Re}(\ln(E(L)))$ rather than $E(L)$.

\item Regarding the plots of $-\text{Re}(\ln E)$, I suspect that the horizontal axis represents a quantity like $1/L$ raised to a certain power: $L^{-5/2}$ in Figs. 1–4, 7, and 8; $L^{-3/2}$ in Fig. 5; and $L^{-2}$ in Fig. 6. Additionally, in Fig. 5, the label "$L^2$" should be corrected to "$L^{-2}$".

\item The definition of $L_{co}^{(1)}$ (first appearing on page 6, before Eq. (28)) is unclear. Could the authors clarify this?

\item To improve readability, especially in power series expansions such as in Eqs. (17) and (29), I suggest labeling the coefficients by the corresponding power, for example:
$$
c_{5/2}L^{-5/2} + c_{3}L^{-3} + c_{7/2}L^{-7/2}.
$$
This would avoid ambiguity, such as using the same symbol $c_1$ for coefficients of different powers.

\item Page 6: The reasoning behind the selected $\alpha$-ranges for states 1, 2,and 3 is not clear. Why is $\alpha \leq \alpha_{\text{crit}} = 1/2$ considered only for state 1, $\alpha > \alpha_{\text{crit}} = 1/2$ only for state 2, and $\alpha = \alpha_{\text{crit}} = 1/2$ only for state 3?

\item Page 6, before Eq. (23): There is a stray "1" in the line containing $P = \frac{2\pi}{L}$.

\item Section 2.3.5: It is stated that "The imaginary part of the eigenvalue vanishes", which appears to contradict Eq. (32), where the eigenvalue for state 3 is complex.

\item Page 15, before Eq. (72): Please correct the typographical error "number number".
\end{enumerate}

Recommendation

Ask for minor revision

---

## Editorial Decision

resubmitted